# Development of nano-emulsions based on *Ayapana triplinervis* essential oil for the control of *Aedes aegypti* larvae

Alex Bruno Lobato Rodrigues[1]*, Rosany Lopes Martins[2], Érica de Menezes Rabelo[2], Rosana Tomazi[2], Lizandra Lima Santos[2], Lethícia Barreto Brandão[2], Cleidjane Gomes Faustino[2], Ana Luzia Ferreira Farias[2], Cleydson Breno Rodrigues dos Santos[2], Patrick de Castro Cantuária[3], Allan Kardec Ribeiro Galardo[4], Sheylla Susan Moreira da Silva de Almeida[1,2]

1 Department of Exact and Technological Sciences, Federal University of Amapá, Macapá, Amapá, Brazil,
2 Department of Biological and Health Sciences, Federal University of Amapa, Macapá, Amapá, Brazil,
3 Amapaense Herbarium, Institute of Scientific and Technological Research of the State of Amapá, Macapá, Amapá, Brazil, 4 Laboratory of Medical Entomology, Institute of Scientific and Technological Research of the State of Amapá, Macapá, Amapá, Brazil

☯ These authors contributed equally to this work.
* alexrodrigues.quim@gmail.com

**Data Availability Statement:** All relevant data are within the paper and its Supporting information files.

## Abstract

*Ayapana triplinervis* is a plant species used in traditional medicine and in mystical-religious rituals by traditional communities in the Amazon. The aim of this study are to develop a nano-emulsion containing essential oil from *A. triplinervis* morphotypes, to evaluate larvicidal activity against *Aedes aegypti* and acute oral toxicity in Swiss albino mice (*Mus musculus*). The essential oils were extracted by steam dragging, identified by gas chromatography coupled to mass spectrometry, and nano-emulsions were prepared using the low energy method. Phytochemical analyses indicated the major compounds, expressed as area percentage, β-Caryophyllene (45.93%) and Thymohydroquinone Dimethyl Ether (32.93%) in morphotype A; and Thymohydroquinone Dimethyl Ether (84.53%) was found in morphotype B. Morphotype A essential oil nano-emulsion showed a particle size of 101.400 ± 0.971 nm (polydispersity index = 0.124 ± 0.009 and zeta potential = -19.300 ± 0.787 mV). Morphotype B essential oil nano-emulsion had a particle size of 104.567 ± 0.416 nm (polydispersity index = 0.168 ± 0.016 and zeta potential = -27.700 ± 1.307 mV). Histomorphological analyses showed the presence of inflammatory cells in the liver of animals treated with morphotype A essential oil nano-emulsion (MAEON) and morphotype B essential oil nano-emulsion (MBEON). Congestion and the presence of transudate with leukocyte infiltration in the lung of animals treated with MAEON were observed. The nano-emulsions containing essential oils of *A. triplinervis* morphotypes showed an effective nanobiotechnological product in the chemical control of *A. aegypti* larvae with minimal toxicological action for non-target mammals.

**Funding:** This study received support from the following sources: The Amapá Research Support Foundation (FAPEAP), to the Research Program for SUS - PPSUS - Ministry of Health (grant number 250,203,026 / 2016); and The Coordination for the Improvement of Higher Education Personnel (CAPES) / Ministry of Education (MEC; process number 1).

**Competing interests:** The authors have declared that no competing interests exist.

## Introduction

*Aedes aegypti* is an anthropophilic insect vector of contagious infections such as Dengue (DENV), chikungunya (CHIKV), Zika (ZIKV) and Yellow Fever in tropical and subtropical regions of the planet [1]. The spread and survival of the species in human clusters is the result of the destruction of its natural environment, wich selected in the wild population the most lantent anthropophilic characteristics of the genus [2, 3].

Another point that requires attention is the double and triple co-infections between DENGV, ZIKV and CHIKV which can be transmitted simultaneously by *A. aegypti* [4]. Therefore, the chemical control applied in the larval phase of *A. aegypti* through the use of plant insecticides is an important tool in stopping mosquito-borne infections and their impacts on public health [5].

Although they are sessile organisms, plants have different forms of defense against phytopathogens and herbivores, including a complex chemical mechanism of secondary metabolites that can be used for chemical control of *A. aegypti*, as the essential oils of the *Ayapana triplinervis* [6].

Originally from South America, *A. triplinervis* (synonym: *Eupatorium triplinerves*) can be found in Brazil, Ecuador, Peru, Puerto Rico and Guyana, besides being adapted in other countries like India and Vietnam [7]. The species is found in Brazil in two morphotypes: Japana-Branca (morphotype A) and Japana-roxa (morphotype B) [8].

*Ayapana triplinervis* alkaloids and tannins exhibit greater food deterrence in *Plutella xylostella* and *Crocidolomia binotalis* agricultural pests, followed by their phenols and flavonoids. *A. triplinervis* extracts disrupted the growth and development of *Myzus persicae* nymphs and showed significant pest control properties, and plant species may be indicated as potential candidates for further study of their potential as botanical pesticides—an alternative to synthetic insecticides [9].

Despite the significant potential for chemical insect control, secondary metabolites from *A. triplinervis*, especially essential oil (EO), are susceptible to volatility and oxidation by oxygen and light, affecting the biological activity of their chemical constituents, or their limited condition of solubility of the essential oil in the aqueous medium, where *A. aegyti* larvae develop [10].

One way to enable protection against degradation and oxidation, and even improve the solubility in aqueous medium of EO is through the development of a technological nanoemulsification system, which offers advantages such as ease of handling, stability, protection against oxidation, better distribution, solubility and controlled release [11].

Nano-emulsions are colloidal systems where their particles are submicron size which are carriers of active molecules. Nanocapsulation technology of sparingly soluble bioactives in aqueous medium is useful in the pharmaceutical and food industries to be applied as bioactivity protection and controlled release to improve bioavailability of active compounds [12]. Nanoformulations represent an emerging technology with application in several areas ranging from the pharmaceutical industry, agriculture, to the control of insects of epidemiological interest for public health. Features such as increased efficiency, durability and reduced effective concentration can bring significant benefits and add value to the nanostructured product [13, 14].

When the particle size distribution of an emulsion is below 80 nm, nano-emulsion has advanced properties compared to conventional sized emulsions as transparent visual appearance, high colloidal stability and a large interfacial area relative to volume [15]. This study hypothesized that nano-emulsions containing *A. triplinervis* essential oil may have insecticidal activity against *A. aegypti* larvae. Therefore, the aims of this study were to evaluate the

larvicidal activity against *A. aegypti* of nano-emulsions containing essential oils of *A. triplinervis* morphotypes and their acute oral toxicity in non-target organism.

## Material and methods

### Plant material

The leaves of *Ayapana triplinervis* morphotypes A and B were collected in the District of Fazendinha (S 0˚ 03'69.55" and W 51˚ 11'03.77"), in Macapá, Amapá. Specimen samples were identified by Doctor Patrick Cantuária and deposited in Herbarium Amapaense (HAMAB) of the Institute of Scientific and Technological Research of Amapá (IEPA) under the codes ABLR001-HAMAB e ABLR002-HAMAB. Access to the plant samples was authorized by the Brazilian Genetic Heritage Management Council under registration number AC90152.

### Essential oils extraction

The essential oils were extracted by hydrodistillation using Clevenger apparatus (São Paulo, SP, Brazil). Dry leaves of morphotype A (2,652 Kg) and morphotype B (0,653 Kg) were extracted for two hours at 100 ˚C. The essential oils were stored in an amber bottle at -4 ˚C under light protection [16].

### Gas chromatography/mass spectrometry analysis

The chemical compositions of the essential oils were determined by gas chromatography coupled to mass spectrometer (GC-MS) [17] on Shimadzu equipment, model CGMS-QP 2010 (Kyoto, Japan), on a DB-5HT column of the J & W Scientific brand, column 30 m, diameter 0.32 mm, film thickness 0.10 μm and nitrogen as carrier gas. The apparatus was operated under internal column pressure of 56.7 kPa, split ratio 1:20, gas flow in the column of 1.0 mL. min$^{-1}$ (210 ˚C), injector temperature 220 ˚C, and the detector (CG-EM) of 240 ˚C. The initial temperature of the column was 60 ˚C with an increase of 3 ˚C.min$^{-1}$ until it reached 240 ˚C, kept constant for 30 minutes. The mass spectrometer was programmed to perform readings in a range of 29 to 400 Da, at intervals of 0.5 seconds, with ionization energy of 70 eV. It was injected 1μL of each sample dissolved in hexane.

A standard mixture of n-alkanes (Sigma-Aldrich, St Louis, MO, C8—C40) was used to verify the performance of the GC-MS system and calculate the linear retention index (LRI) of each compound in the sample. 1 μL of n-alkanes was injected into the GC-MS system operated under the conditions described above, and their respective retention times ($R_T$) were used as an external reference standard for calculating the LRI, together with $R_T$ of each compound of interest. The LRI was calculated according to the literature indication and the phytochemical identification was based on the comparison of their pattern of fragmentation mass spectrum with the library of the equipment and by comparing the calculated linear retention index (LRI) to the literature [18].

### Nano-emulsification method

Nano-emulsions were prepared by low-energy method by phase inversion emulsification according to the methodology of Ostertard et al. [19] and Oliveira et al. [20], which involves the titration of an aqueous phase in an organic phase with constant stirring at room temperature. The organic phase was formed by adding the surfactant to the oil and homogenized for 30 minutes at 750 rpm. Then the aqueous phase was added dropwise and subjected to a magnetic stirrer (model AP59—Phoenix, Araraquara, SP, Brazil) for 60 minutes. The final essential oil concentration was 25,000 μg.mL$^{-1}$ and the surfactant/oil ratio (SOR) equal 1:1.

## Required hydrophilic-lipophilic balance

A mixture of non-ionic surfactants with variable hydrophilic-lipophilic equilibrium value (sorbitan monooleate HLB = 4.3 (Sigma-Aldrich, St Louis, MO), polyoxyethylene sorbitan monooleate HLB = 15 (Sigma-Aldrich, St Louis, MO), polyoxyethylene sorbitan monolaurate HLB = 16.7 (Sigma-Aldrich, St Louis, MO) was used to achieve a range of values of HLB between 8 and 16. The rHLB value of the mixture was calculated according to Oliveira and co-workers [21].

## Particle size distribution and zeta potential measurements

Photon Correlation Spectroscopy analysis was performed using Zetasizer Nano Zs (Malvern Instruments, Malvern, UK) equipment with red laser 10 mW (X = 632.8 nm). Nano-emulsions were diluted in deionized water (1:25, v/v) and analyzed on the 90˚ angle. Droplet Size, polydispersity index, zeta potential were evaluated in triplicate and expressed as mean and standard deviation [21].

## Evaluation of larvicidal lctivity against *A. aegypti*

*Aedes aegypti* larvae (Rockefeller strain, weight 22.6 ± 0.8 mg and leight 4.19 ± 0.49 mm) from the colony of Institute of Scientific and Technological Research of Amapá –IEPA were maintained in a room (3 m x 4 m) with controlled climatic conditions: temperature of 25 ± 2 ˚C, relative air humidity of 75 ± 5%, photoperiod of 12 hours.

The methodology was carried out in accordance with the standardized protocol of the World Health Organizations [22] replacing disposable cups with beakers (120 mL). A preliminary test was conducted at 500, 400, 300, 200 and 100 µg.mL$^{-1}$ in 100 mL of aqueous solution of essential oil with DMSO at 5% or nano-emulsion. The larvicidal evaluation was performed in triplicate for each concentration, then were added 25 *A. aegypti* third-instar larvae and their mortality was assessed at 24 and 48 hours after exposure.

The larvae showed different susceptibility to morphotypes A and B of the species and after a preliminary bioassay, the larvicidal activity of morphotype A essential oil and nano-emulsion were adjusted to 150, 125, 100, 75 e 50 µg.mL$^{-1}$, while morphotype B essential oil and nano-emulsion were adjusted to 100, 80, 60, 40 e 20 µg.mL$^{-1}$. Negative control was 5% dimethyl sulfoxide (DMSO) aqueous solution for the essential oil, and nano-emulsion was used surfactant blenda at 2.5%. The positive control was *Bacillus thuringiensis* subspecies *israelensis* (BTI) bacteria in commercial solution at 0.37 µg.mL$^{-1}$. The tests were carried out in quintuplicates and the average temperature of the aqueous solution was 25˚C.

Immediately after coming into contact with the nano-emulsion, the behavior of the larvae was observed. After 24 and 48 hours of exposure, dead larvae were counted, taking as dead all those unable to reach the surface. Tests in which the negative control mortality exceeded 20% were discarded and repeated, those with mortality between 5 and 20% were corrected using the Abbott's equation [23].

## Light microscopy analysis

Larvae treated in the highest concentration of each nano-emulsion were fixed in 70% ethanol and analyzed under a light microscope model BX41 (Olympus, Tokyo) and photographed in a camera model MDCE 5C according to Botas and co-workers [24].

## Evaluation of acute toxicity of *A. triplinervis* morphotypes nano-emulsions

This study was approved by the Research Ethics Committee of Universidade Federal do Amapá (CEP—Unifap– 004/2019). All procedures were performed in accordance with international animal care protocols and national animal testing regulations. The experiments used male and female, 12-week old Swiss albino mice (*Mus musculus*), provided by the Multidisciplinary Center for Biological Research in the Area of Science in Laboratory Animals (CEMIB). Swiss albino mice (*mus musculus*) represent a model of animal experimentation internationally that can bring relevant information about the toxicological effect in mammals that are not the target of insecticides [25].

Animals were kept in polyethylene cages at a controlled temperature (25 °C) over a 12-hour period of light or dark cycle and had free access to food and water. The animals had restricted food for three hours before the beginning of the experiment, the food was replaced one hour after the treatment with nano-emulsion.

Acute oral toxicity of the nano-emulsions was performed with 03 experimental groups (morphotype A essential oil nano-emulsion (MAEON), morphotyne B essential oil nano-emulsion (MBEON) and negative control), each group with 06 animals (03 males and 03 females).

The test was performed according to the Guideline 423 of the Organization for Economic Co-operation and Development [25] for acute toxic dose class testing. The nanoformulations and negative control were diluted in water at 2,000 mg.kg$^{-1}$ (limit of 1 mL of the solution per 100 g of body weight of the animals) and administered orally through an appropriate cannula.

The Swiss albino mice (*Mus musculus*) were observed at 30, 60, 120, 240 and 360 minutes after oral treatment and daily for 14 days. Behavioural changes were assessed and recorded: general activity, vocal frenzy, irritability, touch response, response to tail stimulus, contortion, posterior train position, straightening reflex, body tone, grip strength, ataxia, auricular reflex, corneal reflex, tremors, convulsions, anaesthesia, lacrimation, ptosis, urination, defecation, piloerection, hypothermia, breathing, cyanosis, hyperaemia and death. Water consumption (mL) and food intake (g) were evaluated daily, and body weight every 03 days.

The animals were sacrificed on the 14th in a $CO_2$ chamber, obeying the ethical principles of animal experimentation. Liver, heart, kidneys and lungs were removed for weighing and macroscopic and histopathological analysis [26].

## Histopathological analysis

Liver, heart, kidneys and lungs were fixed in 10% formalin buffer solution for 48 hours, then they were dehydrated in different concentrations of alcohols (70%, 80%, 90%, and 100%) and diaphanized in xylol and embedded in paraffin. Paraffin sections (5 μm) were stained with haematoxylin and eosin. The slides were examined under a light microscope model BX41 (Olympus, Tokyo) and the magnified images of the tissues' structure were captured for further study [26].

## Statistical analysis

The data were organized into mean and standard deviations. Probit analyses were used to determine the larvae's susceptibility and ANOVA to determine the statistical difference between treatments using the Statistical Package for the Social Sciences program (version 22, Chicago, Illinois), with a 95% confidence limit.

**Table 1. Chemical composition of essential oils of *A. triplinervis* morphotypes A and B.**

| Compound | Morphotype A | | Morphotype B | | LRI [18] |
|---|---|---|---|---|---|
| | % Peak Area | LRI | Percentege | LRI | |
| α-Pinene | 0.93 | 948 | 1.25 | 948 | 932 |
| β-Pinene | 2.16 | 943 | 2.26 | 943 | 974 |
| Thymol Methyl Ether | 0.68 | 1231 | - | - | 1232 |
| α-Gurjunene | 0.98 | 1419 | 0.41 | 1419 | 1409 |
| β-Caryophyllene | 45.93 | 1494 | - | - | 1419 |
| Thymohydroquinone Dimethyl Ether | 32.93 | 1423 | 84.53 | 1423 | 1424 |
| α-Humulene | 1.60 | 1579 | 1.11 | 1579 | 1452 |
| Aciphyllene | 0.53 | 1490 | - | - | 1501 |
| β-Selinene | 0.88 | 1469 | 0.41 | 1469 | 1489 |
| α-Muurulene | 0.45 | 1435 | - | - | 1392 |
| Valencene | - | - | 0.53 | 1474 | 1496 |
| γ-Gurjenene | 0.64 | 1461 | - | - | 1475 |
| Caryophyllene Oxide | 0.76 | 1506 | 0.67 | 1507 | 1582 |
| 2,5-Di-Tert-Buthyl-1,4-Benzenoquinone | 4.22 | 1633 | 1.82 | 1633 | - |
| α-Cedrene Epoxide | - | - | 0.93 | 1293 | 1574 |

LRI = Linear retention index calculated; LRI [18] = Linear retention index of literature.

## Results

The morphotype A essential oil showed a translucent colour and yield 0.248% ± 0.037 (m/m). Phytochemical analyses of the Morphotype A essential oil (MAEO) by GC-MS indicated the major compounds β-Caryophyllene (45.93%) and Thymohydroquinone Dimethyl Ether (32.93%). The Morphotype B essential oil (MBEO) yield 0.753% ± 0.467 m/m and was purple, Thymohydroquinone Dimethyl Ether (84.53%) was the only compound in greater quantity identified in the MBEO by GC-MS.

Thymol Methyl Ether, γ-Gurjunene and β-Caryophyllene were exclusively detected in morphotype A, and Valencene and α-Cedrene Epoxide were found only in morphotype B, as shown in Table 1. Additional phytochemical data is available on S1 and S2 Figs and S1 and S2 Tables.

The content of the α-Pinene and β-Pinene monoterpenes added in each morphotype was equivalent to 3.09% in the MAEO and 3.51% in the MBEO. Oxygenated derivatives from *p*-Cymene such as Thymohydroquinone Dimethyl Ether and 2,5-Di-Tert-Buthyl-1,4-Benzene-quinone were found in morphotypes A and B, while Thymol Methyl Ether was found only in morphotype A. Oxygenated derivatives of p-cymene totalled 37.00% in morphotype A and 86.35% in morphotype B.

Reports in the literature have indicated that the *Eupatoruim capillifolium* essential oil, of which contain 57.1% oxygenated derivatives from p-cymene (36.3% Thymol Methyl Ether and 20.8% Thymohydroquinone Dimethyl Ether), presented repellent activity against *A. aegypti* and adulticide activity against *Stephanitis pyrioides* [27].

Carbocyclic and oxygenated sesquiterpenes were found in greater quantity and diversity in morphotype A than in morphotype B. The sesquiterpene β-Caryophyllene was predominant only in morphotype A (45.93%). Scientific literature has reported that antioxidant effect of β-caryophyllene protects rat liver from carbon tetrachloride-induced fibrosis [28].

Studies have shown a variation in the chemical composition of the essential oil of *A. triplinervis*. However, Thymohydroquinone Dimethyl Ether has been found in high concentrations.

Specimens from Lagoa Grande, Amapá State (Brazil) reported the presence of the majority compounds Thymohydroquinone Dimethyl Ether with 69.7% and β-Caryophyllene with 19.7% [29]. Another study that analysed the phytochemical composition of *A. triplinervis* essential oil collected on Reunion Island indicated the presence of Thymohydroquinone Dimethyl Ether at 89.9% to 92.8% [7]. However, none of the studies differentiated the chemical composition of the essential oils of each *A. triplinervis* morphotype.

Some species of the Asteraceae family produce Thymohydroquinone Dimethyl Ether as major compounds, among them *Eupatorium marginatum* (64.2%) [30], *Laggera alata* (44.0%) [31], *Pulicania mauritanica* (37.2%) [32], *Laggera pterodonta* (30.5%) [33], *Blumea perrotteti-ana* (30,0%) [34] and *Sphaeranthus indicus* (18.2%) [35].

Asteraceae are important for their recognized insecticidal activity described in the literature [36, 37]. Alkalamides and coumarins showed insecticidal activity against *Sitophylus oryzae* and *Rhyzopertha dominica* [38], polyacetylenes and terpenes have proven insecticide and repellent activities against larvae and adults of *A. aegypti* [39–41].

Essential oils are good candidates for insecticides, however, their terpenes have low solubility in aqueous media, which represents the first stage of development of culicids such as *A. aegypti*, and are subject to degradation by oxidation and volatility [42]. To improve the solubility of essential oils in water, nano-emulsions were developed by the low energy method using phase inversion composition. This nanoencapsulation technique has advantages: non-heating, solvent free, low cost, and environmentally friendly [43].

Structures with a particle size ranging from 88,830 nm to 333,600 nm were produced for the morphotype A essential oil nano-emulsions (MAEON), and between 99.637 nm and 279.300 nm for the morphotype B essential oil nano-emsulsion (MBEON) on day 0 of analysis. Nano-emulsions formed with the hydrophilic-lipophilic balance (HLB) between 10 and 15 were homogeneous and milky, where the greater HLB's (15.5 and 16) the nano-emulsions showed a bluish reflex for both morphotypes, as seen in Fig 1 and S3 and S4 Figs.

Light scattering on nanoscale colloidal systems causes a characteristic bluish reflection called the Tyndall Effect and represents a parameter indicative of the kinetic stability acquired by nano-emulsions compared to macroemulsions [44]. However, nano-emulsion are susceptible to flocculation, coalescence and sedimentation instabilities [45]. Table 2 shows the physical

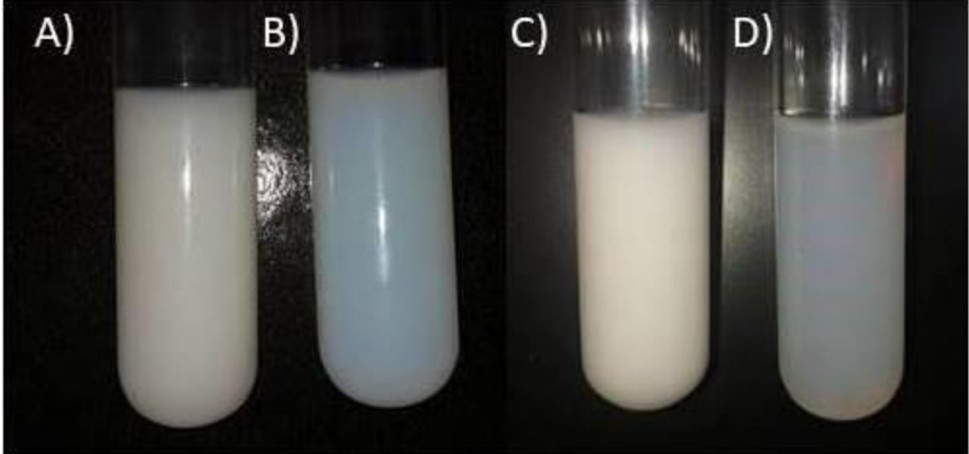

**Fig 1. *A. triplinervis* morphotype A nano-emulsion in A) HLB 14 and B) HLB 16, and morphotype B nano-emulsion in C) HLB 14 e D) 16 on day 0.**

**Table 2. Reports of the most promising and optimized nano-emulsion (HLB 16) of each *A. triplinervis* morphotype over time.**

| Nano-emulsion | Time (day) | HLB | Particle Size (nn) | Polydispersity Index | Zeta Potential (mV) | Electrical Conductivity (mS.cm$^{-1}$) |
|---|---|---|---|---|---|---|
| MAEON | 0 | 16 | 88.830 ± 0.948 | 0.138 ± 0.012 | -23.200 ± 0.458 | 0.025 ± 0.011 |
| | 7 | 16 | 92.783 ±0.225 | 0.120 ± 0.007 | -17.667 ±0.404 | 0.021 ±0.004 |
| | 14 | 16 | 101.167 ± 0.971 | 0.125 ± 0.009 | -20.100 ± 2.787 | 0.021 ± 0.004 |
| | 21 | 16 | 99.463 ± 0.242 | 0.121 ± 0.008 | -32.767 ± 1.450 | 0.014 ± 0.002 |
| MBEON | 0 | 16 | 99.637 ± 0.529 | 0.213 ± 0.011 | -22.000 ± 1.153 | 0.020 ± 0.013 |
| | 7 | 16 | 104.067 ± 0.306 | 0.186 ± 0.012 | -22.400 ± 1.153 | 0.022 ± 0.001 |
| | 14 | 16 | 106.567 ± 0.416 | 0.168 ± 0.016 | -26.700 ± 1.308 | 0.013 ± 0.006 |
| | 21 | 16 | 99.420 ± 0.519 | 0.132 ±0.008 | -31.033 ± 2.178 | 0.020 ± 0.013 |

MAEON = Morphotype A Essential Oil Nano-emulsion. MBEON = Morphotype B Essential Oil Nano-emulsion; HLB = hydrophilic-lipophilic balance value

parameters of the most promising nano-emulsions of the essential oil of each *A. triplinervis* morphotype.

The narrowest emulsions were produced in HLB 16, and their particle size ranged from 88.830 nm to 99.463 nm for the Morphotype A essential oil, and between 99.637 nm to 99.420 nm for the Morphotype B essential oil between days 0 and 21, as shown in the particle size distribution in Fig 2. Emulsions with particle diameters between 20 and 200 nm are called nano-emulsions and possess stability against cremation, flocculation, and sedimentation, but are susceptible to Ostwald ripening over time [45, 46].

The photon correlation spectroscopy analyses indicated that the morphotype A essential oil nano-emulsion (MAEON) had a particle diameter equal to 101.400 ± 0.971 nm (PdI = 0.124 ± 0.009 and ZP = -19.300 ± 0.787 mV). Morphotype B essential oil nano-emulsion (MBEON)

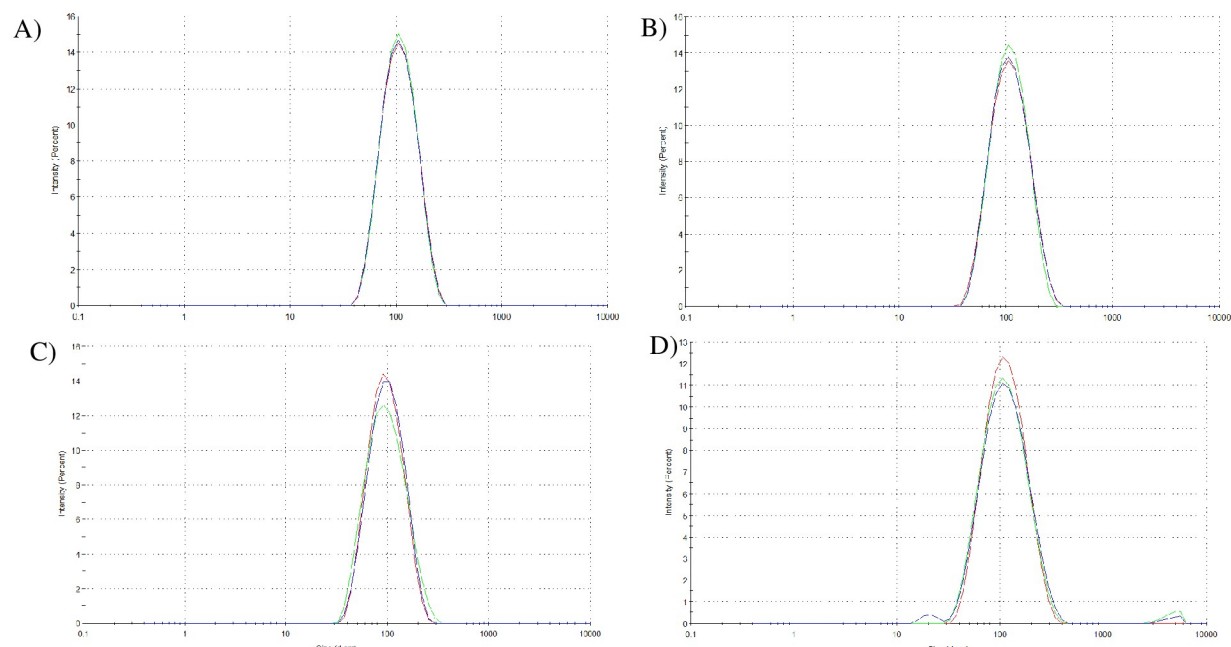

**Fig 2. Particle size distribution of the *A. triplinervis* essential oils nano-emulsion A) morphotype A on day 0; B) morphotype B on day 0; C) morphotype A on day 21; and D) morphotype B on day 21.** The X axis has the particle diameter (nm) and the Y axis has the peak intensity (%).

showed particle size 104.567 ± 0.416 nm (PdI = 0.168 ± 0.016 and ZP = -27.700 ± 1.307 mV), as indicated in the particle size distribution and zeta potential distribution graph in Fig 3.

The solubility of a nano-emulsion can be assessed in terms of the electrical conductivity resulting from the residues of the surfactant organic groups [47]. MAEON presented electrical conductivity equal to 0.021 ± 0.004 mS.cm$^{-1}$, and MBEON showed conductivity of 0.022 ± 0.001 mS.cm$^{-1}$. These results indicate that MAEON and MBEON are soluble in aqueous media [48]. Therefore, it can be concluded that the nano-emulsions containing the essential oils of *A. triplinervis* morphotypes present stability and solubility, and can be used for chemical control of *A. aegypti* larvae.

In insecticidal activity, *A. aegypti* larvae showed atypical behaviour ranging from excitation to lethargy when in contact with essential oil or nano-emulsions, and their mortalities varied as a function of exposure time. Larvae of *A. aegypti* exposed to compounds with neuromuscular action have symptoms such as excitation, convulsions, paralysis, and death [49]. Symptoms observed in the experiment suggest toxic neuromuscular effects from nerve synapses through inhibition of octopamine [50].

Analysis of variance showed differences in the distribution of mean larval mortality caused by nano-emulsions compared to mortality from bulk essential oils (p-value = 0.7340 e F = 0.6247) and indicate that the larvae were more susceptible to nano-emulsions, as shown in Tables 3 and 4.

Nanoencapsulated system provides essential oil with chemical stability, solubility and bio-availability [51]. Bioavailability is facilitated by passive transport of nano-emulsions into cells when they are less than 200 nn in diameter and may represent an increase in nano-emulsion insecticidal activity [52].

Larvicidal compounds may act by absorption into epithelial tissue, respiratory or ingestion or systemic effect on diffusion by different tissues [53]. The mortality effect of *A. aegypti* larvae in contact with nano-emulsions and bulk essential oils as a function of time can be visualized by lethal concentrations that kills 50% (LC$_{50}$) and 90% (LC$_{90}$) exposed larvae in Table 5.

Nano-emulsions show toxicity action that could be observed in larval morphology. The magnitude of toxicity was assessed by a combination of confocal and scanning electron

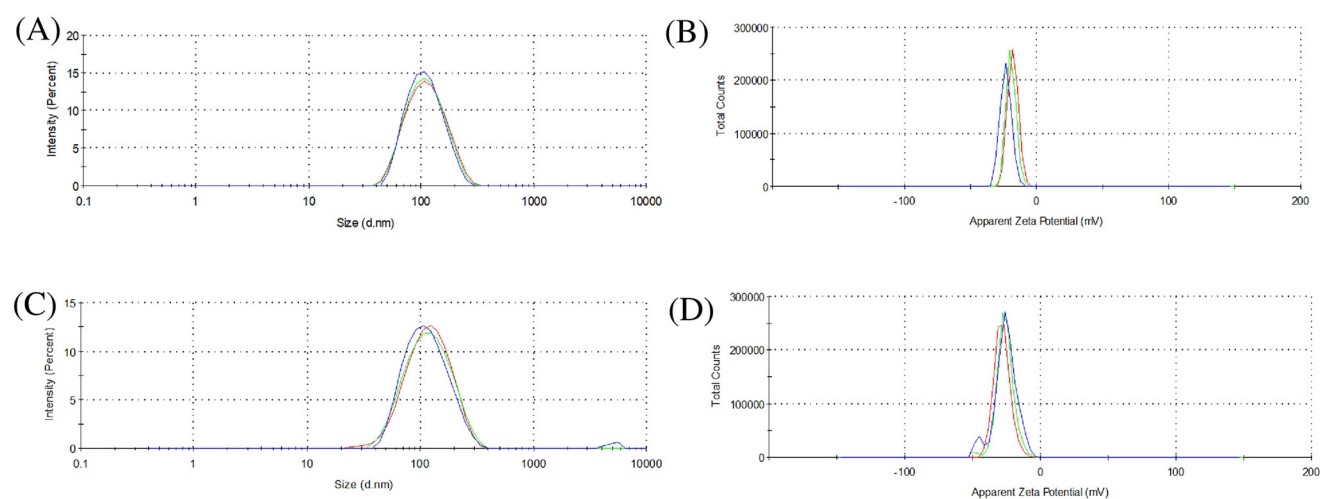

**Fig 3. Size distribution by intensity and zeta potential distribution of nano-emulsion containing essential oil of *A. triplinervis* morphotype A (A and B) and morphotype B (C and D).** The X axis has the particle diameter (nm) and the Y axis has the peak intensity (%).

**Table 3. Mortality of *A. aegypti* larvae in different concentrations of essential oil and nano-emulsion of *A. triplinervis* morphotype A.**

| Concentration (µg.mL$^{-1}$) | MAEO | | MAOEN | |
|---|---|---|---|---|
| | Mortality (%) | | Mortality (%) | |
| | 24 h• | 48 h• | 24 h* | 48 h* |
| 150 | 84.80 ± 8.67 [a] | 91.20 ± 8.67 [a] | 92.80 ± 3.35 [a] | 96.80 ± 1.79 [a] |
| 125 | 37.60 ± 12.84 [ab] | 47.20 ± 12.46 [a] | 64.00 ± 29.53 [a] | 76.00 ± 38.78 [a] |
| 100 | 25.60 ± 12.84 [ac] | 39.20 ± 15.34 [a] | 35.20 ± 20.86 [ab] | 39.20 ± 11.10 [a] |
| 75 | 12.80 ± 7.69 [a] | 20.00 ± 5.66 [a] | 27.20 ± 8.20 [ac] | 29.60 ± 7.27 [ab] |
| 50 | 10.40 ± 6.07 [a] | 16.80 ± 5.93 [a] | 22.40 ± 8.29 [a] | 27.20 ± 12.13 [ac] |
| Negative Control | 0.00 ± 0.00 [a] | 0.00 ± 0.00 [a] | 0.00 ± 0.00 [a] | 0.00 ± 0.00 [a] |
| Positive Control | 100 ± 0.00 [a] | 100 ± 0.00 [a] | 100 ± 0.00 [a] | 100 ± 0.00 [a] |

Means within a column with different letters are significantly different. Means within a row with different simbols are significantly different (p-value ˂0.001, F = 13,464).

**Table 4. Mortality of *A. aegypti* larvae in different concentrations of essential oil and nano-emulsion of *A. triplinervis* morphotype B.**

| Concentration (µg.mL$^{-1}$) | MBEO | | MBEON | |
|---|---|---|---|---|
| | 24 h• | 48 h• | 24 h* | 48 h* |
| 100 | 60.00 ± 11.31[a] | 64.80 ± 10.35[a] | 88.80 ± 5.22[a] | 92.80 ± 5.22[a] |
| 80 | 42.40 ± 8.29[ab] | 49.60 ± 8.76[ab] | 80.00 ± 6.93[a] | 81.60 ± 8.29[a] |
| 60 | 30.40 ± 6.07[ac] | 40.80 ± 8.67[ac] | 65.60 ± 15.65[a] | 67.20 ± 20.28[a] |
| 40 | 13.60 ± 5.37[a] | 22.40 ± 7.80[a] | 54.40 ± 33.42[ba] | 43.20 ± 24.23[ba] |
| 20 | 0.80 ± 1.79[a] | 3.20 ± 3.35[a] | 5.60 ± 2.19[ba] | 34.40 ± 28.93[ba] |
| Negative Control | 0.00 ± 0.00[a] | 0.00 ± 0.00[a] | 0.00 ± 0.00[ba] | 0.00 ± 0.00[ba] |
| Positive Control | 100 ± 0.00 [a] | 100 ± 0.00 [a] | 100 ± 0.00 [a] | 100 ± 0.00 [a] |

Means within a column with different letters are significantly different. Means within a row with different simblos are significantly different (p-value = 0.001, F = 7,984).

microscopy. Results indicated deformation of anal papillae, and changes deriving from peeling epithelial tissue in the region located in the of the larvae abdomen, as shown in Fig 4.

Essential oils are highly selective for octopaminergic receptors that are distributed in neural and non-neural tissues [54]. Studies have shown that essential oils can modify the activity of

**Table 5. Lethal concentration in order to kills 50% and 90% of 3$^{rd}$ instar larvae of *A. aegypti* in contact with the essential oils or the nano-emulsions of *A. triplinervis* morphotypes, and their lower and upper limits.**

| | 24 h | | 48 h | |
|---|---|---|---|---|
| | LC$_{50}$ (µg.mL$^{-1}$) | LC$_{90}$ (µg.mL$^{-1}$) | LC$_{50}$ (µg.mL$^{-1}$) | LC$_{90}$ (µg.mL$^{-1}$) |
| MAEO | 122.082 | 235.379 | 105.979 | 218.231 |
| | (99.630–181.737) | (165.731–872.543) | (84.020–146.794) | (154.044–778.960) |
| | $X^2$ (df) = 9,975 (3) | | $X^2$ (df) = 3,667 (3) | |
| MAEON | 96.228 | 202.042 | 87.434 | 177.207 |
| | (73.912–127.785) | (144.733–675.615) | (65.953–110.646) | (131.862–455.481) |
| | $X^2$ (df) = 3,535 (3) | | $X^2$ (df) = 4,430 (3) | |
| MBEO | 86.196 | 200.901 | 75.347 | 202.062 |
| | (65.812–177.738) | (122.421–2351.066) | (55.442–135.702) | (119.379–2097.984) |
| | $X^2$ (df) = 0.091 (3) | | $X^2$ (df) = 0.123 (3) | |
| MBEON | 44.755 | 100.473 | 35.570 | 117.450 |
| | (30.681–57.919) | (73.997–214.623) | (16.191–50.818) | (74.818–629.081) |
| | $X^2$ (df) = 0.826 (3) | | $X^2$ (df) = 1,269 (3) | |

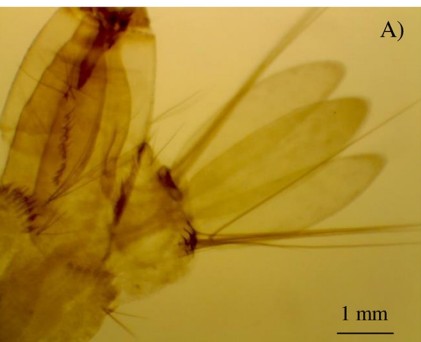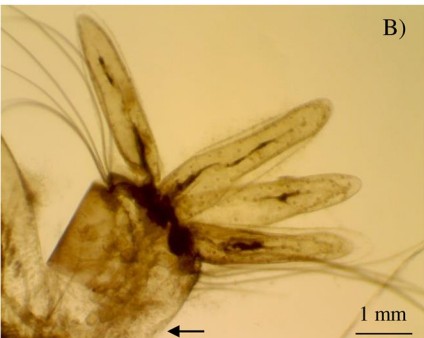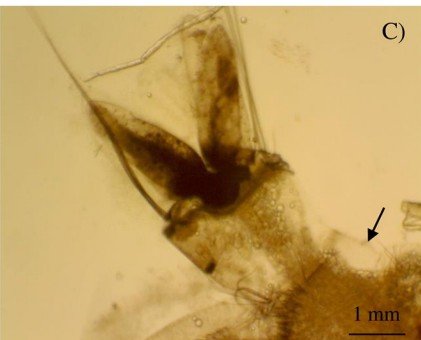

**Fig 4. Morphological changes in the anal papillae of *A. aegypti* 3<sup>rd</sup> instar larvae in contact with nano-emulsions of morphopotype A essential oil (B) and morphotype B (C) of *A. triplinervis* compared to control (A) by confocal microscopy (25×).**

neurons by octopamine receptors [55–57]. Authors suggest that octopamine acts on insect epithelial tissue to provide plasticity for abdominal distension in breathing and larval movement [58].

Loss of epithelial tissue impairs larval mobility and respiration. Anal papillae morphology and surface dysfunctions affect regulation of osmotic functions and, together with cuticle loss, represent a condition that is detrimental to the survival of *A. aegypti* larvae. [49]. These data are in agreement with the results observed for *Pterodon emarginatus* nano-emulsion in *Culex quinquefasciatus* larvae [21].

In assessment of acute oral toxicity in non-target mammals, mortality was not observed in 2000 mg.Kg$^{-1}$ of MAEON and MBEON during the 14 days of observation. Hippocratic screening showed that MAEON caused irritability in males and females during the first hour, and corneal reflex in females during the first 3 hours of observation. MBEON caused writhing and irritability in males and females during the first hour of analysis, as shown in Table 6.

In physiological analysis, a decrease in water intake was observed in males and females of the MAEON group, and an increase in water consumption was observed for males and females of the MBEON group. However, these results did not present significant differences when compared with the control group. No changes in food intake were observed in the MAEON and MBEON groups, and the body weight gain did not show significant differences when compared to the control group, as shown in Table 7.

There were no significant differences in relative weight of heart, kidney, and lung when compared to control group organs. However, there was a statistically significant decrease in the absolute liver weight of males treated with MBEON compared to the control group, as shown in Table 8.

**Table 6. Effect of oral administration of MAEON and MBEON (2000 mg.kg$^{-1}$) on behavioural parameters of Swiss albino mice (*Mus musculus*) during acute toxicity assessment.**

| Group | Adults | N° of mice | N° of dead mice | Toxicity Symptoms |
|---|---|---|---|---|
| Control | Male | 3 | 0 | - |
| | Female | 3 | 0 | - |
| MAEON | Male | 3 | 0 | Irritability |
| | Female | 3 | 0 | Irritability and corneal reflex |
| MBEON | Male | 3 | 0 | Contortion and irritability |
| | Female | 3 | 0 | Contortion and irritability |

**Table 7. Physiological parameters of Swiss albino mice (*Mus musculus*) treated with MAEON and MBEON at 2000 mg.kg⁻¹.**

| Parameters | Control | | MAEON | | MBEON | |
|---|---|---|---|---|---|---|
| | **Male** | **Female** | **Male** | **Female** | **Male** | **Female** |
| Water (mL) | 13.16 ± 1.46 | 13.35 ± 1.46 | 10.27 ± 2.19 | 12.96 ± 1.68 | 15.34 ± 1.08 | 15.92 ± 1.26 |
| Food (g) | 6.39 ± 0.34 | 6.30 ± 0.25 | 6.24 ± 0.51 | 5.81 ± 0.46 | 6.16 ± 1.33 | 5.91 ± 0.34 |
| Body Weight (g) | 28.07 ± 1.45 | 28.85 ± 1.33 | 29.50 ± 1.44 | 25.55 ± 1.07 | 31.47 ± 0.37 | 24.75 ± 0.39 |

Values are expressed as mean ± standard deviation (n = 3). Statistical significance was calculated using One-way Anova followed by Bonferroni's multiple comparison test (p <0.01).

Histological evaluation showed no changes in the liver, heart, lung and kidneys of the control group. However, inflammatory cells were observed in the liver of animals treated with MAEON and MBEON. In the lungs, there was congestion and the presence of transudate with leukocyte infiltration in animals treated with MAEON. The hearts and kidneys of these animals were not affected, as shown in the Fig 5. Histopathological analysis of the liver identified a discrete leukocyte infiltrate in the sample. The leukocyte infiltrate may be associated with an increase in the structure of the organ and, as a consequence, an increase in the weight of the liver of the animals treated with MBEON in comparison with the control. Because it is the organ responsible for the metabolism of chemicals and the regulation of the immune response, the liver becomes susceptible to toxic damage and may undergo morphological changes, as indicated [59, 60].

## Discussion

Few studies are dedicated to reporting the biological activity of *A. triplinervis* in the literature: anxiolytic, antinociceptive and sedative effect on the central nervous system [61], inhibitory activity in the formation of melanin in B16 melanoma cells [62], inhibition of the tyrosinase enzyme [63], antimicrobian activity [64, 65]. Essential oil and Thymohydroquinone Dimethyl Ether isolated from the species demonstrated potential inhibition of ZIKV infection in human cells [66].

The first available chemical data of the *A. triplinervis* in the scientific literature were reported by Semmler in 1908 and indicated 80% of Thymohydroquinone Dimethyl Ether [67]. We report in this study, for the first time in the scientific literature, the phytochemical composition of the essential oils of *A. triplinervis* morphotypes A and B.

Two distinct chemotypes of Thymohydroquinone Dimethyl Ether have been cited in the literature: one in Lucknow (India), which presented Selin-4 (15), 7 (11) -dien-8-one with 36.6%; and the second was found in Tamil Nadu (India), which indicated the presence of

**Table 8. Effect of MAEON and MBEON at 2000 mg.kg⁻¹ on the weight of different Swiss albino (*Mus musculus*) organs after 14 days of treatment.**

| Organs (%) | Control | | MAEON | | MBEON | |
|---|---|---|---|---|---|---|
| | **Males** | **Females** | **Males** | **Females** | **Males** | **Females** |
| Liver | 5.97 ± 0.42 | 6.27 ± 0.28 | 7.55 ± 1.32 | 7.16 ± 0.50 | 6.49 ± 0.25* | 6.18 ± 0.43 |
| Kidneys | 0.81 ± 0.12 | 0.84 ± 0.12 | 0.99 ± 0.16 | 0.76 ± 0.03 | 0.88 ± 0.05 | 0.77 ± 0.04 |
| Heart | 0.58 ± 0.07 | 0.54 ± 0.04 | 0.55 ± 0.09 | 0.56 ± 0.14 | 0.49 ± 0.02 | 0.48 ± 0.01 |
| Lungs | 0.73 ± 0.02 | 0.74 ± 0.02 | 0.88 ± 0.11 | 0.85 ± 0.15 | 0.74 ± 0.15 | 0.65 ± 0.15 |

Values are expressed as mean ± standard deviation (n = 3). Statistical significance was calculated using One-way Anova followed by Bonferroni's multiple comparison test (p <0.01).

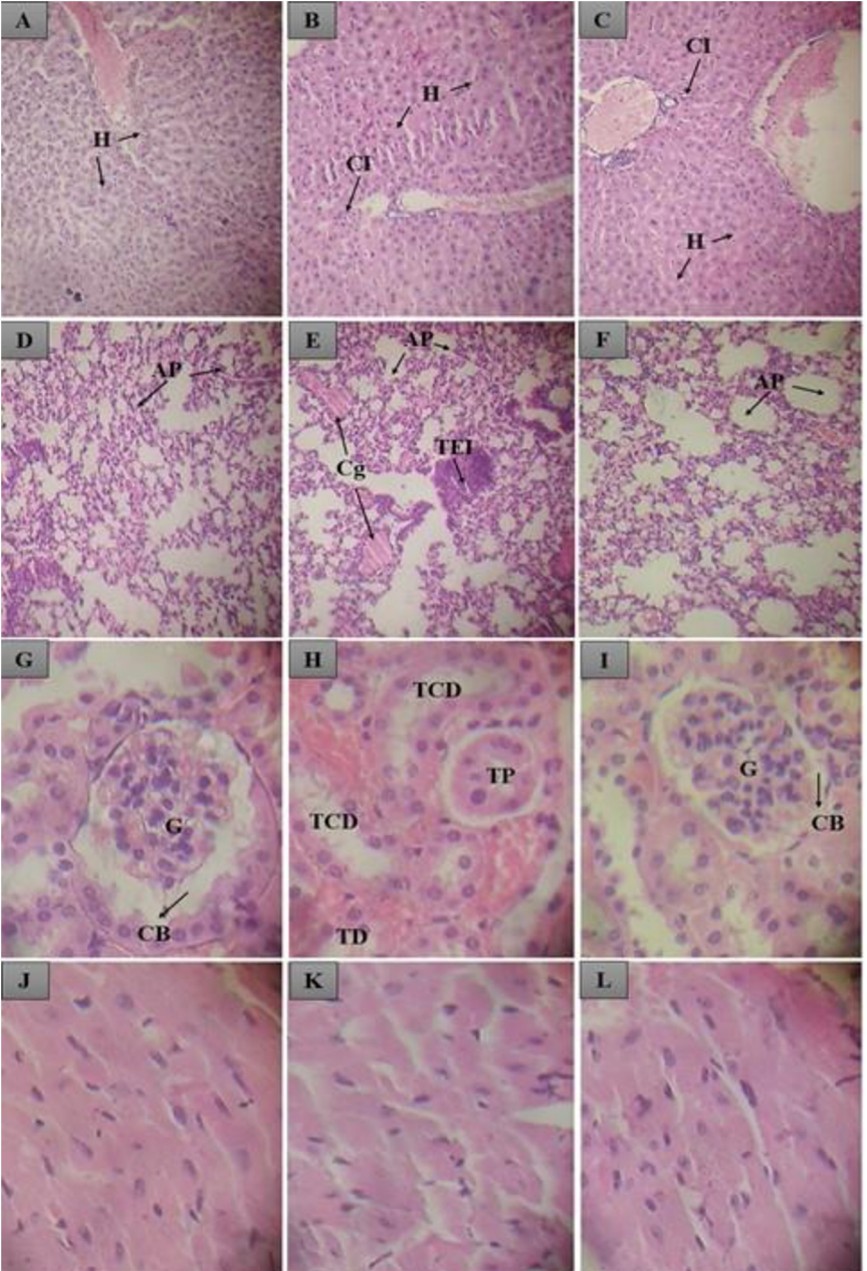

**Fig 5. Histological section of animals treated orally with control and MAEON and MBEON.** In A, the normal liver of a control group animal; In B and C, the liver of an animal treated with MAEON and MBEON, with normal hepatocytes (H) and the presence of inflammatory cells (CI). In D and F, the normal lungs of control and MBEON animals, where pulmonary alveoli was observed; In E, the lung of a MAEON group animal, where pulmonary alveoli (AP), congestion (Cg) and oedematous transudate with leukocyte infiltration (TEI) were observed. In G, H and I, the normal kidneys of control, MAEON and MBEON groups with glomerulus (G), Bowmam capsule (CB), distal tubule (DT), proximal tubule (TB). In J, K and L, normal hearts are observed from the control group, MAEON and MBEON.

2-Tert-Butyl-1,4-dimethoxybenzene with a percentage of 74.27% [68, 69]. However, no study has differentiated the chemical composition of the essential oil of each morphotype.

The interspecific variability of essential oils produces different chemotypes by a combination of genetic and environmental factors [70] and may contribute to the diversification of ecological interactions between plants and insects [71, 72].

*A. triplinervis* morphotype A essential oil presented Thymohydroquinone Dimethyl Ether and β-caryophyllene as major compounds. β-caryophyllene is a sesqueterpene that has a wide distribution in species of the family Astereaceae and may have anti-inflammatory, antibiotic, insecticide, antioxidant and anticarcinogenic activity [73]. In morphotype B, the concentration of Thymohydroquinone remained similar to that found in our previous studies. The chemical composition can vary through climatic and environmental changes [74], and these results can support the understanding of chemical variability in the species and its morphotypes, however, further studies are needed.

The essential oil of morphotype A is the closest to the chemotype described in Brazil by Maia and his co-workers [29], while the essential oil of morphotype B has a greater chemotypic relationship with the specimen described on Reunion Island by Gauvin-Bialacki and Maradon [7]. A different chemotype was reported in India and indicated Selina-4(15),7(11)-dien-8-one as a major compound with a percentage of 36,6%, followed by β-Caryophyllene with 14.7% [68].

The scientific literature indicates that essential oils are considered eco-friendly insecticides [69], act in low concentrations at the active site, and remain for a short time exposed in nature, however, when inserted into a nanostructured system, they show improvements in insecticidal activity, stability, and safety in their application due to specificity selectivity in insects compared to humans [70].

Essential oils are susceptible to deterioration caused by temperature and radiation, which can induce oxidation of their chemical components and cause significant changes in their pharmacological activity [71]. Therefore, it was possible to develop nano-emulsion as a drug delivery system using low energy emulsification method through composition inversion, which uses low-energy input without heating and is solvent-free organic [20] to improve the chemical stability of *A. triplinervis* essential oils [70].

Nano-emulsions can be prepared by the high energy method that uses mechanical devices capable of creating shear stress for formation of finely distributed droplets, and the low-energy method, which depends on spontaneous emulsification within the oil-surfactant-water mixture, altered by changes in phase inversion system or solvent separation [72].

Our studies have demonstrated eco-friendly, non-heated and solvent-free nano-emulsions that improved the insecticidal activity against *A. aegyti* in relation to bulk essential oils with $LC_{50}$ < 100 μg.mL$^{-1}$ in 24 hours of exposure. This result corroborates other studies published in the literature that showed that the nano-emulsion of *Vitex negundo* is more effective in controlling *A. aegypti* larvae than its bulk essential oil [73], and that the nano-emulsion of *Anethum graveolens* is more toxic to *Anopheles stephensi* than its bulk essential oil [74].

The *A. aegypti* larvae were more susceptible to *A. triplinervis* nano-emulsions than *Baccharis reticularia* (Asteraceae) nano-emulsion with particle size equal to 92.9 nm (PdI = 0.412 e ZP = -20.4) and $LC_{50}$ = 221.273 μg.mL$^{-1}$ [26]. The larvicidal activity of *A. triplinervis* morphotypes nano-emulsion was more pronounced than *Copaifera duckei* nano-emulsion (PS = 145.2 nm and PdI = 0.378) that showed 70% mortality at 200 μg.mL$^{-1}$ [47], and *Pterodon emarginatus* nano-emulsion (PS = 128.0 nm and PdI = 0.250) that demonstrated $LC_{50}$ = 371.6 μg.mL$^{-1}$ in *A. aegypti* larvae [75].

When the particle size distribution is polymodal, i.e with a Polydispersity Index (PdI) greater than 0.3, nano-emulsions are susceptible to Ostwald ripening degradation. The mechanism of instability is through the chemical potential difference between smaller droplets and larger droplets caused by mass transport between them, and can cause degradation by coalescence or flocculation [76, 77]. Polydispersity index of morphotype A and B nano-emulsions demonstrated uniform particle size and monomodal distribution, thus is stable against Ostwald ripening [78, 79].

Zeta potential comes from the surface distribution of charges in nano-emulsions and its value of around ± 30 mV indicates a stable system through electrostatic repulsion that prevents flocculation degradation in nano-emulsions [80, 81]. From a zeta potential standpoint, the NOEMA and NOEMB acquired stability over time. The low solubility of essential oils makes their application in chemical control of *Aedes aegypti* larvae unfeasible, therefore, an O/W (Oil-in-Water) nano-emulsion adds advantages as a nanobiotechnological product because it improves solubility of essential oils in aqueous media [82].

The literature reports some cases that use the low-energy non-heat method to form an insecticidal nano-emulsions-based delivery system with chemical and kinetic stability in HLB 15. *Baccharis reticular* essential oil, which had a particle size of 138 nm (PdI = 0.453 and ZP = -18.3 mV), and *P. emarginatus* essential oil nano-emulsion with particle size 116.8 nm (PdI = 0.250 and ZP = -19.9 mV) [24, 83].

In addition, *Rosmarinus officinalis* essential oil nano-emulsions were produced in HLB 16.7 with a particle size of 174.1 nm (PdI = 0.136 and ZP was not reported) [82]. The contextualization of these results indicates that the nano-emulsion droplet diameter prepared by the phase inversion method depends on the number of HLB for formulating colloidal stability systems [84].

In this study we also exclusively report colloidal systems with high stability in HLB 16 for the essential oils of *A. triplinervis* morphotypes, which presented monomodal distribution nanostructure and narrow hydrodynamic diameters produced by the low-energy method, non-heat, solvent-free and environmentally friendly with low production costs [20].

The $LC_{50}$ and $LC_{90}$ were lower at 48 hours of exposure compared to 24 hours for essential oil and nano-emulsion of morphotype A. The variation in larvicidal activity over time may be due to the increased concentration of bioactive compounds that are released over time in a nanoformulation [85]. These results are in line with previous studies showing that micro or nano-emulsions are more effective in chemical control of *Culex pipiens pipiens*, *Culex quinquefasciatus* and *Anopheles stephensi* larvae compared to their bulk essential oils [74, 86, 87].

Comparing the $LC_{50}$ between nano-emulsion and the bulk essential oil of each morphotype, it can be observed that the insecticidal activity of morphotype B was more effective in the mortality of *A. aegypti* larvae than morphotype A. The most pronounced insecticidal bioactivity of morphotype B may be due to the higher percentage of Thymohydroquinone Dimethyl Ether or by the synergism of all phytochemical constituents.

Contrary to this study, the literature reports that *Eupatorium capillifolium* essential oil, which contains 20.8% Thymohydroquinone Dimethyl Ether, has moderate adulticidal and larvicidal activity but high repellent activity against *A. gambiae* [88].

*Laggera alata* essential oil contains 24.4% of Thymohydroquinone Dimethyl Ether and has been described with $LC_{50}$ = 273.38 mg.mL$^{-1}$ for *A. aegypti* [89]. *Guateria hispida* essential oils were characterized with 21.0% β-caryophyllene, and *Piper arboreum* essential oil with 10.88% showed $LC_{50}$ of 85.74 μg.mL$^{-1}$ and 55.00 μg.mL$^{-1}$ agains *A. aegypti* larvae [90, 91]. The results obtained in this study compared to literature data may indicate that the larvicidal activity is related to the major constituents or through the synergistic action of the secondary metabolites [92].

The evaluation of larvicidal activity shows that the essential oil of morphotype A did not show larvicidal activity in *A. aegypti* ($LC_{50}$ > 100 μg.mL$^{-1}$), however, the nano-emulsion of morphotype A and the essential oil of morphotype B were active in the control of *A. aegypti* larvae ($LC_{50}$ < 100 μg.mL$^{-1}$). Morphotype B essential oil nano-emulsion was the most effective in the chemical control of *A. aegypti* larvae ($LC_{50}$ < 50 μg.mL$^{-1}$) [93].

Therefore, it can be concluded that the nano-emulsions containing the essential oils of *A. triplinervis* morphotypes A and B showed nanobiotechnological potential to be applied in the chemical control of *A. aegypti* larvae in preliminary laboratory tests.

The larvicidal activity of *A. triplinervis* essential oils may indicate neuromuscular action by inhibiting the octapaminergic system [94]. Octopamine is a neuromodulator, neurotransmitter and neurohormone that acts on the insect's nervous system and represents a specific marker of insecticidal activity in essential oils [95].

From the morphological point of view, deformations in the anal papillae influence the larval osmoregulation ability and motility [96]. Similar changes in shrinking of the internal cuticle of anal papillae have been described in *A. aegypti* larvae in contact with aqueous extracts of different species of the genus Piper and *Argemone mexicana* [97].

The depreciating effects caused by MAEON and MBEON in the anal papillae and in the cuticle affect motility and influence the survival of *A. aegypti* larvae. Similar effects are observed in the morphology of *Culex quinquefaciatus* larvae by the action of *P. emarginatus* nano-emulsion and *A. aegypti* larvae by the action of *B. reticulate* nano-emulsion and may indicate the toxicity mechanism of *A. triplinervis* essential oil nano-emulsions [21, 24].

The development of formulations for the controlled release of terpenes has great potential for reducing side effects and promoting prolonged insecticidal action [98]. Due to the small size that favours rapid absorption and metabolism, the nano-emulsions have the potential to maximize the toxic effect at high concentrations [99], as shown in this study.

Contortion, irritability and corneal reflex indicate action in the central nervous system, and these observations were in accordance with previous studies that demonstrate sedative, anxiolytic and antidepressant effects in different animal models for *A. triplinervis* extract [84].

The oral bioavailability of nano-emulsions increases as a function of the particle size, and its surface charge can influence the potential for acute toxicity through the solubility balance, lipid digestion rate and mucosal penetration rate [100].

The presence of leukocyte infiltrates in the liver, congestion in the alveolar septa and transudate with leukocyte infiltration in the lungs indicate inflammatory processes in these organs caused by MAEON and MBEON [101]. However, the literature has suggested that the application of insecticidal nano-emulsion in non-target mammals is safe [102].

Insecticidal nano-emulsion containing *Eucalyptus staigeriana* essential oil did not show haematological changes in rats [103]. In another study, a nano-emulsion prepared from *Manilkara subsericea* did not show death or toxicological effects on mice [104]. Nano-emulsion containing *Carapa guianensis* did not show cytotoxicity, genotoxicity and hepatotoxicity in mice [105], and Thymohydroquinone Dimethyl Ether, the major metabolite of the samples, showed no toxicity in Zebrafish model [66].

The evaluation of acute toxicity showed that the results found for *A. triplinervis* essential oils nano-emulsions are in accordance with the results described in the literature. The absence of mortality in the nano-emulsion concentration used in this study suggests that the $LD_{50}$ is superior to the treatment applied (2000 mg.Kg$^{-1}$) and indicates that nano-emulsions have low toxicity and can be classified in toxicity category 5 [97].

## Conclusions

The results indicated that morphotype A presented a different phytochemical composition to that of the major compounds from the morphotype B essential oil. The essential oil nano-emulsions demonstrated kinetic stability, aqueous solubility, monomodal distribution and particle size 101.400 ± 0.971 for morphotype A, and 104.567 ± 0.416 for morphotype B.

The evaluation of larvicidal activity showed that *A. aegypti* larvae were more susceptible to nano-emulsions than to bulk essential oil. The magnitude of the deleterious effects of nano-emulsions in larvae occurred through the deformation of the anal papillae and the peeling of the cuticle when compared with the control group.

The assessment of acute oral toxicity indicated action on the central nervous system, and histopathological analysis showed damage to the lungs and livers of the animals, however, there was no mortality at the tested dose, indicating that nano-emulsions can be classified in category 5 of toxicity.

In this context, *A. triplinervis* morphotypes essential oils nano-emulsions proved to be an eco-friendly nanobiotechnological product, effective for chemical control of *A. aegypti* larvae and safe for non-target mammals that can be used to control of diseases DENGV, CHIKV, ZIKV.

## Supporting information

**S1 Fig. Chromatogram of the essential oil of *A. triplinervis* morphotype A.**
(DOCX)

**S2 Fig. Chromatogram of the essential oil of *A. triplinervis* morphotype B.**
(DOCX)

**S3 Fig. Physical parameters of *A. triplinervis* morphotype A nano-emulsions in different hydrophilic-lipophilic balance values at 0, 7, 14 and 21 days.**
(DOCX)

**S4 Fig. Physical parameters of nano-emulsions *A. triplinervis* morphotype B nano-emulsions in different hydrophyl-lipophyle balance values at 0, 7, 14 and 21 days.**
(DOCX)

**S1 Table. Chemical composition, retention time (RT), percentage and calculated Linear Retention Index (LRI) and literature of essential oil of *A. triplinervis* morphotype A.**
(DOCX)

**S2 Table. Chemical composition, retention time (RT), percentage and calculated Linear Retention Index (LRI) and literature of essential oil of *A. triplinervis* morphotype B.**
(DOCX)

## Acknowledgments

To the Pharmaceutical Research laboratory—UNIFAP under the responsibility of Dr. José Carlos Tavares Carvalho. To the Laboratory of Toxicology and Pharmaceutical Chemistry—UNIFAP under the responsibility of Dr. Mayara Amoras Teles Fujishima. The Dean of Research and Graduate Studies—PROPESPG—UNIFAP.

## Author Contributions

**Conceptualization:** Alex Bruno Lobato Rodrigues, Sheylla Susan Moreira da Silva de Almeida.

**Data curation:** Alex Bruno Lobato Rodrigues.

**Formal analysis:** Alex Bruno Lobato Rodrigues.

**Investigation:** Alex Bruno Lobato Rodrigues, Rosany Lopes Martins, Érica de Menezes Rabelo, Rosana Tomazi, Lizandra Lima Santos, Lethícia Barreto Brandão, Cleidjane Gomes

Faustino, Ana Luzia Ferreira Farias, Cleydson Breno Rodrigues dos Santos, Patrick de Castro Cantuária, Allan Kardec Ribeiro Galardo.

**Project administration:** Sheylla Susan Moreira da Silva de Almeida.

**Supervision:** Sheylla Susan Moreira da Silva de Almeida.

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
