## [Decision Letter · Decision Letter 0]

11 Jan 2021

PONE-D-20-35553

Development of nano-emulsions based on Ayapana triplinervis essential oil for the control of Aedes aegypti larvae

PLOS ONE

Dear Dr. de Almeida,

Thank you for submitting your manuscript to PLOS ONE. After careful consideration, we feel that it has merit but does not fully meet PLOS ONE’s publication criteria as it currently stands. Therefore, we invite you to submit a revised version of the manuscript that addresses the points raised during the review process.

We look forward to receiving your revised manuscript.

Kind regards,

Ahmed Ibrahim Hasaballah

Academic Editor

PLOS ONE

Journal Requirements:

2) In your Methods section, please provide additional information regarding the permits you obtained for the work. Please ensure you have included the full name of the authority that approved the plant material collection site access and, if no permits were required, a brief statement explaining why.

Reviewers' comments:

Reviewer's Responses to Questions

**Comments to the Author**

1. Is the manuscript technically sound, and do the data support the conclusions?

Reviewer #1: Partly

Reviewer #2: Yes

2. Has the statistical analysis been performed appropriately and rigorously? 

Reviewer #1: Yes

Reviewer #2: Yes

3. Have the authors made all data underlying the findings in their manuscript fully available?

Reviewer #1: Yes

Reviewer #2: Yes

4. Is the manuscript presented in an intelligible fashion and written in standard English?

Reviewer #1: No

Reviewer #2: Yes

5. Review Comments to the Author

Reviewer #1: The manuscript by Rodrigues et al describes the activity of extracts from two morphotypes of Ayapana triplinervis on Aedes aegypti larvae and in mice. Therein they provide physical characteristics of the extracted essential oils and nano-emulsions prepared from them. The methods that were employed are standard for the field and conducted appropriately. The principal findings include: (1) detailed descriptions of the physical and chemical properties of the extracts, (2) bioassays showing that larval mortality increases with concentration of the extracts and that the nano-emulsions are generally more effective, and (3) low toxicity when orally administered to mice.

While the organization of the manuscript was reasonably linear, it contained some grammatical errors and was unfocused in some sections. The latter was most evident in the Introduction and Discussion. Nearly a third of the Introduction describes the diseased that are transmitted by Ae. aegypti, but these diseases are not part of the study. Additionally, the Results section contained discussion material and some data was mentioned but the results not described (e.g. Supplementary Data). If the data is not described in the Results or Discussion, it should be removed from the manuscript. Of a minor note, it appears that some words should be translated to English. Overall, I recommend the authors refocus the writing and strive to reduce the length of the manuscript. Doing so should improve the impact of the work substantially.

Notably, the Supplementary Data contains mass spectra chromatograms that appear to have been published elsewhere (e.g., by NIST and Adams (2017)). I doubt that PLOS permits the republishing of figures. I recommend the authors remove those figures and either replace them with mass spectra of standards that they generate for each chemical that was analyzed or state the diagnostic fragment ions of each chemical they analyze that match what is reported in the literature.

Below are some comments on specific line numbers in the manuscript should be also be addressed if a revision is submitted.

Mice were used in the study, thus an Ethics Statement is required by PLOS.

16-17: Two symbols are used to indicate equal contribution and each author has one or the other. This suggests that all authors contributed equally and only one symbol is needed.

26: not necessary to capitalize Gas Chromatography or Mass Spectrometry

27-30: I don’t think it necessary to cite the protocols used for larval bioassay and toxicity in the abstract.

31-33: Define what % is referring to (assuming it’s of the total but is it by mass, volume, etc).

34: PdI and ZP are non-standard acronyms and may need to be defined for readers to know their meaning.

47-67: Extensive introduction of DENV, CHIKV and ZIKV is provided but are not highly relevant to the study. Recommend reducing the extent of the introduction of these topics to simply state they are transmitted by Ae. aegypti.

49-51: not accurate that Ae. aegypti spread due to habitat destruction; it is the case they spread because they their habitats were replicated in urban settings and they adapted to it.

77: sentences that start with a species name should not use abbreviation

Methods: all equipment and reagents used in the study should include the manufacturer and location of the company

197: PLOS may require IACUC protocol or approval number to be included in the manuscript.

207: What does “03” refer to?

229: Please describe how the tissues were routinely processed.

230: State microscope brand, model, etc.

239: MAEO is not previously defined

240: Should E be replaced with “Beta”?

256-291: This is discussion that should be moved to Discussion. Note that the Discussion is already overly lengthy.

309: Please state in the Methods how the emulsions were stored for the 21 day study.

Table 2: Only particle size is mentioned in the manuscript outside of this table. The other physical characteristics should be described elsewhere or removed if not of great relevance to the study.

Fig 2 and 3: Fonts for x- and y-axis are too small to view.

333-347: Please integrate into the Discussion

345: O/W (and later, W/O) is not defined

359-360: Please provide data that supports this conclusion.

Table 3 and 4: This data would be much easier to interpret were it presented as a scatter plot with linear regression line fitted to the data. Alternatively, these tables could be removed as the LC50 and LC90 data provided in Table 5 may be sufficient for demonstrating the effect of the extracts on larval mortality. If these tables are retained (not recommended), please state what the dots and asterisk on 24 h and 48 h indicate.

Table 5: Statistical analysis of MAEO vs MAEON and 24h vs 48 h should be conducted.

386: Incorrect statement: LC90 for MBEO and MBEON at 48h was higher than 24h. These differences should be described in the manuscript.

387-415: Discussion material

394-396: There is not sufficient data to draw this conclusion (the most prevalent component may not have the highest biological activity).

397-399: Please indicate how this study contradicts what is presented in the manuscript.

404-406: This statement is not well supported by the data.

408 - 410: Please provide data that supports the rationale for selecting 100 ug/ml as a value for drawing comparisons (it seems arbitrary).

413-415: Field trials of the emulsions would need to be conducted to draw this conclusion.

418: The surface areas of the structures for the controls and treatments should be measured to draw meaningful conclusions that are supported by the data. Samples were fixed in 70% ethanol which would likely deform the structures and reduce their volume. Notably, Figure 4D is indicated as a control, but appears to be more deformed than the treatment displayed in Figure 4E.

419: Please provide arrowheads in Fig4 that show where epithelial tissue is peeling.

Fig 4: Scale bars must be included for each image.

459: Text states there is a significant decrease in liver weight, the weight of the liver of male mice treated with MBEON was higher than control (an increase). Please explain.

Discussion: as noted previously, the Discussion is overly lengthy and relatively unfocused. Many paragraphs are a single run-on sentence. Please revise to improve focus.

523: Please provide evidence that the larvicide is eco-friendly.

622 and elsewhere: The convention is to use commas for separating thousands and periods to signify the decimal place in a value.

Supplementary Tables 7 – 12: These tables are not highly relevant as the results are summarized in Table 5. However, if they are retained, the number of significant digits must be reduced to reflect the actual sample size (mortality was unlikely measured to the thousandths level of precision).

Mass spectra in Supplementary Figures lack label on the x- and y-axis (e.g., m/z, relative abundance, retention time (m))

Tables define LRI Lit. as Linear retention index of literature, which as stated is nonsensical. Recommend replacing with “Linear retention index that is reported in the literature (16))” “16” is the reference.

S15: These images should be analyzed (e.g. structures measured), annotated (e.g. arrowheads) and have scale bars or removed from the manuscript.

Reviewer #2: Review report

Manuscript Number: PONE-D-20-35553

Article Type: Research Article

Full Title: Development of nano-emulsions based on Ayapana triplinervis essential oil for the control of Aedes aegypti larvae

General Comments

Nanopesticides are formulated on the basis of different nanocarrier and the principal chemical targeted for management of insect pests. Application of the insecticides in nature bears inherent problem of affecting the non-target organisms and alteration of the ambient environment. Despite higher efficacy of the insecticides, the cost-effectiveness and the ill-effects to the environment are the primary factors impeding the wide use of the pesticides. Continuous effort in improvement of the pesticides is required to combat the mosquito menace and thus the incidence of mosquito borne diseases. Mosquitoes can exploit wide range of water bodies for breeding, and the strategies for the biological regulation vary accordingly. In majority instances, the preference of the small containers and small spaces as breeding habitats by Aedes aegypti calls for application of the botanicals for effective regulation. No doubt, Ae. aegypti can exploit various other breeding habitats, the application of the botanicals are useful depending its cost-effectiveness, including least effect on non-target organism and reasonable price. Thus the present article is useful from the viewpoint of Ae. aegypti regulation, where the efficacy of a nano-emulsion linked with the essential oil of Ayapana triplinervis was assessed. In general the methodology and results are acceptable, though, improvement in the language quality is required for the article to qualify for publication. Going through the manuscript, I feel that the article is useful and qualifies for publication. However, a revision is required in language quality and the tables and figures to promote the biological control of the dengue vectors with the nano-emulsions. The following points should be considered by the authors during the revision of the manuscript.

Introduction section

1] Please improve the overall language quality.

2] Consider reducing the number of paragraphs and increase clarity in the working hypothesis

3] Provide few statements on the utility of the study and prospective application. I hope you would agree that several botanicals and synthetic pesticides have been highlighted for their efficacy and use in mosquito regulation, particularly, Ae. aegypti and Ae. albopictus. So, please emphasize the significance of the present work in the context of cost and efficacy for Ae. aegypti regulation.

4] Please consider additional references provided for inclusion in the introduction section.

Methodology section

Please provide an experimental outline in a table or through a flow diagram.

Please elaborate further the methodology mentioned in the section “Evaluation of Larvicidal Activity against A. aegypti’. Please highlight the number of replicates (line 180…is understandable but expand further) used for each dose.

Please expand and elaborate the statements from lines 182 through 185.

Provide a logic for using the Swiss albino mice and its significance of the study in the methods section.

Please provide reasons for not using probit analysis for deducing the median dose for the nano pesticides.

In addition, please provide a clarification about how the Ae. aegypti instar III larvae were identified. Mention the origin of the P generation of Ae. aegypti. I think it would be prudent to provide the length and weight measures of the larvae…. Please consider this suggestion. You are suggesting about the most recent method of mosquito control, so please be specific about the larvae to make the whole compilation more logical and attractive for the readers and scientist.

Why mouse was considered as non-target organism and not freshwater organisms? It would have been prudent if the freshwater organisms were considered.

Results

Considering the results shown in Table 3 and Table 4, it appears that these two tables indicate positive control showed 100% mortality against both morphotype A and B. The biopesticide Bacillus thuringiensis subspecies israelensis (BTI) was used as positive control. If positive control showed 100% mortality it can be said that it has better efficacy than that of the experimental combinations. Then what is the logic to build a larvicide which actually have lower efficacy than that of established larvicide also Bacillus thuringiensis don’t have any toxic effect on non targeted animal. So please provide explanation or consider alternative presentation.

Please consider application of probit analysis for the dose mortality data.

Discussion Section

Please consider the following articles and the relevant points as applicable for the article.

Provide a comparative account of the various botanicals and their efficacy against Ae. aegypti and elaborate how the application of Ayapana triplinerviswould be useful.

Please depict the prospective utility of the present study highlighting whether the nano-emulsion bears prospect in terms of cost and the effects.

Balaure PC, Dragoş B, Gudovan D, Gudovan I. 2017. 4 - Nanopesticides: a new paradigm in crop protection. New Pesticides and Soil Sensors, 129-192. https://doi.org/10.1016/B978-0-12-804299-1.00005-9

Chowdhury N, Ghosh A, Chandra G. 2008.Mosquito larvicidal activities of Solanum villosum berry extract against the dengue vector Stegomyia aegypti. BMC Complementary Alternative Medicine, 8, 10 (2008).

Djiwanti SR, Kaushik S. 2019. Nanopesticide: Future Application of Nanomaterials in Plant Protection. In: Prasad R. (eds) Plant Nanobionics. Nanotechnology in the Life Sciences. 255-298 pp.Springer, Cham. https://doi.org/10.1007/978-3-030-16379-2_10

Facknath S, Lalljee B. 2008. Study of various extracts of Ayapana triplinervis for their potential in controlling three insect pests of horticultural crops. Tropicultura, 26(2): 119 – 124.

Ghosh A, Chowdhury N, Chandra G. 2012. Plant extracts as potential mosquito larvicides. Indian Journal of Medical Research, 135:581-598.

Kah M, Kookana RS, Gogos A, Bucheli TD. 2018. A critical evaluation of nanopesticides and nanofertilizers against their conventional analogues. Nature Nanotechnology, 13: 677–684. https://doi.org/10.1038/s41565-018-0131-1

Kookana RS, Boxall ABA, Reeves PT, Ashauer R, Beulke S, Chaudhry Q, Cornelis G, Fernandes TF, Gan J, Kah M, Lynch I, Ranville J, Sinclair C, Spurgeon D, Tiede K, Van den Brink PJ. 2014. Nanopesticides: Guiding Principles for Regulatory Evaluation of Environmental Risks. Journal of Agricultural and Food Chemistry, 62 (19): 4227-4240. DOI: 10.1021/jf500232f

Priyanka P, Kumar D, Yadav K, Yadav A. 2019. Nanopesticides: Synthesis, Formulation and Application in Agriculture. In: Abd-Elsalam K., Prasad R. (eds) Nanobiotechnology Applications in Plant Protection. Nanotechnology in the Life Sciences. 129-143 pp. Springer, Cham. https://doi.org/10.1007/978-3-030-13296-5_7

Rawani A, Chowdhury N, Ghosh A, Laskar S, Chandra G. 2013. Mosquito larvicidal activity of Solanum nigrum berry extracts. Indian J Med Research,137:972-976.

Rawani A, Ghosh A, Chandra G. 2014.Laboratory evaluation of molluscicidal & mosquito larvicidal activities of leaves of Solanum nigrum L. Indian Journal of Medical Research, 140:285-295.

Stadler T, Butelar M, Valdez SR, Gitto JG. 2017. Particulate nanoinsecticides: a new concept in insect pest management. DOI: 10.5772/intechopen.72448

Hope these suggestions are useful in enhancement of the manuscript.

Thank you.

6. PLOS authors have the option to publish the peer review history of their article (what does this mean?). If published, this will include your full peer review and any attached files.

Reviewer #1: No

Reviewer #2: No

---

## [Author Response · Author response to Decision Letter 0]

25 Mar 2021

Ahmed Ibrahim Hasaballah

Academic Editor

PLOS ONE

We are writing to respond to reviewers for the requested adjustments to the manuscript entitled “Development of nano-emulsions based on Ayapana triplinervis for the control of Aedes aegypti larvae”.

JOURNAL REQUIREMENTS:

Response: The style was revised according to the guidance.

2) In your Methods section, please provide additional information regarding the permits you obtained for the work.

Responde: The number of the document authorizing access to the genetic heritage has been added to the methodology.

REVIEWER #1:

• Mice were used in the study, thus an Ethics Statement is required by PLOS.

Response: The approval code of the ethics and animal research committee was provided on lines 197 and 198. Details of Humane endpoints were provided in a specific form of the Journal in the submission.

• 16-17: Two symbols are used to indicate equal contribution and each author has one or the other. This suggests that all authors contributed equally and only one symbol is needed.

Response: adjusted according to the reviewer's request.

• 26: not necessary to capitalize Gas Chromatography or Mass Spectrometry

Response: adjusted according to the reviewer's request.

• 27-30: I don’t think it necessary to cite the protocols used for larval bioassay and toxicity in the abstract.

Response: adjusted according to the reviewer's request.

• 31-33: Define what % is referring to (assuming it’s of the total but is it by mass, volume, etc).

Response: adjusted according to the reviewer's request.

• 34: PdI and ZP are non-standard acronyms and may need to be defined for readers to know their meaning.

Response: adjusted according to the reviewer's request.

• 47-67: Extensive introduction of DENV, CHIKV and ZIKV is provided but are not highly relevant to the study. Recommend reducing the extent of the introduction of these topics to simply state they are transmitted by Ae. aegypti.

Response: adjusted according to the reviewer's request.

• 77: sentences that start with a species name should not use abbreviation

Response: adjusted according to the reviewer's request.

• Methods: all equipment and reagents used in the study should include the manufacturer and location of the company

Response: adjusted according to the reviewer's request.

• 197: PLOS may require IACUC protocol or approval number to be included in the manuscript.

Response: (CEP – Unifap – 004/2019), in line 190, is the approval number by IACUC.

• 207: What does “03” refer to?

Response: Experimental grups: MAEON; MBEON; and Negative Control.

• 229: Please describe how the tissues were routinely processed.

Response: adjusted according to the reviewer's request.

• 230: State microscope brand, model, etc.

Response: adjusted according to the reviewer's request between lines 224 and 226

• 239: MAEO is not previously defined

Response: adjusted according to the reviewer's request.

• 240: Should E be replaced with “Beta”?

Response: adjusted according to the reviewer's request.

• 256-291: This is discussion that should be moved to Discussion. Note that the Discussion is already overly lengthy.

Response: adjusted according to the reviewer's request.

• 309: Please state in the Methods how the emulsions were stored for the 21 day study.

Responde: Lines 159 to 160.

• Table 2: Only particle size is mentioned in the manuscript outside of this table. The other physical characteristics should be described elsewhere or removed if not of great relevance to the study.

Response: We confirm the importance of physical and chemical characterization. Otherwise, we emphasize the contextualization of particle size, polydispersivity index and zeta potential between lines 330 to 350; electrical conductivity between lines 355 to 361. As for electrophoretic mobility, we chose to exclude such data as suggested by the reviewer.

• Fig 2 and 3: Fonts for x- and y-axis are too small to view.

Response: adjusted according to the reviewer's request in lines 321-322; and 332-333

• 333-347: Please integrate into the Discussion

• Response: adjusted according to the reviewer's request.

• 345: O/W (and later, W/O) is not defined

Response: adjusted according to the reviewer's request.

• 359-360: Please provide data that supports this conclusion.

Response: adjusted according to the reviewer's request.

• Table 3 and 4: This data would be much easier to interpret were it presented as a scatter plot with linear regression line fitted to the data. Alternatively, these tables could be removed as the LC50 and LC90 data provided in Table 5 may be sufficient for demonstrating the effect of the extracts on larval mortality. If these tables are retained (not recommended), please state what the dots and asterisk on 24 h and 48 h indicate.

Response: Dots and asterisks indicate statistical differences on the same line. The meaning is in lines 367 and 368 below the column.

• Table 5: Statistical analysis of MAEO vs MAEON and 24h vs 48 h should be conducted. 

Response: adjusted according to the reviewer's request.

• 386: Incorrect statement: LC90 for MBEO and MBEON at 48h was higher than 24h. These differences should be described in the manuscript.

Response: adjusted according to the reviewer's request.

• 394-396: There is not sufficient data to draw this conclusion (the most prevalent component may not have the highest biological activity).

Response: adjusted according to the reviewer's request.

• 397-399: Please indicate how this study contradicts what is presented in the manuscript.

Response: The opposite condition is related to the high content of Thimohydroquinone Dimethyl Ether in morphotype B that showed high larvicidal activity, different from the studies indicated in the reference that presented moderate larvicidal and adulticidal activity of Eupatorium capillifolium that presents with Thymohydroquinone Dimethyl Ether as one of the major compound.

• 404-406: This statement is not well supported by the data.

Response: The Declaration has been corrected.

• 408 - 410: Please provide data that supports the rationale for selecting 100 ug/ml as a value for drawing comparisons (it seems arbitrary).

Response: The LC50 comparison is not arbitrary and is in accordance with the classification proposed by Cheng et al (2003), as mentioned in the paragraph.

• 413-415: Field trials of the emulsions would need to be conducted to draw this conclusion.

Response: The Declaration has been corrected.

• 418: The surface areas of the structures for the controls and treatments should be measured to draw meaningful conclusions that are supported by the data. Samples were fixed in 70% ethanol which would likely deform the structures and reduce their volume. Notably, Figure 4D is indicated as a control, but appears to be more deformed than the treatment displayed in Figure 4E.

Response: The item was adjusted according to what was requested. SEM analyzes were removed to avoid compromising the paper. The result was described according to the new context

• 419: Please provide arrowheads in Fig4 that show where epithelial tissue is peeling.

Response: adjusted according to the reviewer's request.

• Fig 4: Scale bars must be included for each image.

Response: adjusted according to the reviewer's request.

• 459: Text states there is a significant decrease in liver weight, the weight of the liver of male mice treated with MBEON was higher than control (an increase). Please explain.

Response: adjusted according to the reviewer's request. The explanation has been added to the text on lines 425-432 .

• Discussion: as noted previously, the Discussion is overly lengthy and relatively unfocused. Many paragraphs are a single run-on sentence. Please revise to improve focus.

Response: adjusted according to the reviewer's request. Some paragraphs have been deleted

• 523: Please provide evidence that the larvicide is eco-friendly.

Response: A reference was added to the item to support the conclusion.

• 622 and elsewhere: The convention is to use commas for separating thousands and periods to signify the decimal place in a value.

Response: adjusted according to the reviewer's request.

• Supplementary Tables 7 – 12: These tables are not highly relevant as the results are summarized in Table 5. However, if they are retained, the number of significant digits must be reduced to reflect the actual sample size (mortality was unlikely measured to the thousandths level of precision).

Response: The tables were removed according to the reviewer's request.

• S15: These images should be analyzed (e.g. structures measured), annotated (e.g. arrowheads) and have scale bars or removed from the manuscript.

Response: The figures were removed according to the reviewer's request.

REVIEWER #2:

Introduction section 

1] Please improve the overall language quality. 

Response: The tables were removed according to the reviewer's request.

2] Consider reducing the number of paragraphs and increase clarity in the working hypothesis 3] Provide few statements on the utility of the study and prospective application. I hope you would agree that several botanicals and synthetic pesticides have been highlighted for their efficacy and use in mosquito regulation, particularly, Ae. aegypti and Ae. albopictus. So, please emphasize the significance of the present work in the context of cost and efficacy for Ae. aegypti regulation. 

Response: 

Response: adjusted according to the reviewer's request.

4] Please consider additional references provided for inclusion in the introduction section

• The following references have been added to the manuscript:Kookana RS, Boxall AB, Reeves PT, Ashauer R, Beulke S, Chaudhry Q, et al. Nanopesticides: guiding principles for regulatory evaluation of environmental risks. J Agric Food Chem. 2014; 62: 4227-4240. doi: 10.1021/jf500232f

• Kah M, Kookana RS, Gogos A, Bucheli, TD. A critical evaluation of nanopesticides and nanofertilizers against their conventional analogues. Nat Nanotechnol. 2018, 13: 677-684. doi: 10.1038/s41565-018-0131-1

• Silvério MRS, Espindola LS, Lopes NP, Vieira PC. Plant natural products for the control of Aedes aegypti: The main vector of important arboviruses. Molecule. 2020, 25: 3484 - 3526. doi: 10.3390/molecules25153484

• Badawi MS. Histological study of the protective role of ginger on piroxicam-induced liver toxicity in mice. J Chin Med Assoc. 2019, 82:11-18. doi: 10.1016/j.jcma.2018.06.006

• Yuan L, Kaplowitz N. Mechanisms of drug-induced liver injury. Clin Liver Dis. 2013, 17: 507-518. doi: 10.1016/j.cld.2013.07.002

Please elaborate further the methodology mentioned in the section “Evaluation of Larvicidal Activity against A. aegypti’. Please highlight the number of replicates (line 180…is understandable but expand further) used for each dose

Please provide reasons for not using probit analysis for deducing the median dose for the nano pesticides.

Response: Probit analysis was used to deduct the LC50 for essential oil and nanoemulsion of each morphotype of the plant species, as shown in Table 5.

In addition, please provide a clarification about how the Ae. aegypti instar III larvae were identified. Mention the origin of the P generation of Ae. aegypti. I think it would be prudent to provide the length and weight measures of the larvae…. Please consider this suggestion. You are suggesting about the most recent method of mosquito control, so please be specific about the larvae to make the whole compilation more logical and attractive for the readers and scientist.

Response: The information was added on lines 157-158

Why mouse was considered as non-target organism and not freshwater organisms? It would have been prudent if the freshwater organisms were considered.

Response: Mice were considered in this study as non-target organisms due to the anthropophilic characteristics of Ae aegypti. When developing in urban breeders, the direct application of a nanoinsecticide can be ingested through the consumption of water by different domestic animals and even humans. However, future studies may be conducted to assess toxicity in freshwater organisms.

Results 

Considering the results shown in Table 3 and Table 4, it appears that these two tables indicate positive control showed 100% mortality against both morphotype A and B. The biopesticide Bacillus thuringiensis subspecies israelensis (BTI) was used as positive control. If positive control showed 100% mortality it can be said that it has better efficacy than that of the experimental combinations. Then what is the logic to build a larvicide which actually have lower efficacy than that of established larvicide also Bacillus thuringiensis don’t have any toxic effect on non targeted animal. So please provide explanation or consider alternative presentation. Please consider application of probit analysis for the dose mortality data.

Response: BTI, as a positive control, was used as a parameter to compare the magnitude of larvicidal activity of A. triplinervis nanoformulations. Despite showing a higher percentage, BTI alone is not able to effectively supply all public policies for the control of Aedes aegypti. In this sense, the nanoformulations of A. triplinervis can be inserted as an auxiliary integrative practice in the control of Ae aegypti. Allied to this point, this study has additional value when it demonstrates the phytochemical profile of essential oils of the species' morphotypes exclusively in the scientific literature. Therefore, we consider A. triplinervis nanoformulations with scientific and biological value within the proposal presented.

Discussion Section 

Please consider the following articles and the relevant points as applicable for the article. Provide a comparative account of the various botanicals and their efficacy against Ae. aegypti and elaborate how the application of Ayapana triplinerviswould be useful. Please depict the prospective utility of the present study highlighting whether the nano-emulsion bears prospect in terms of cost and the effects.

Response: We reorganized the paragraphs to meet the requested demand and inserted some of the references to clarify the ideas.

Thank you for your time and consideration to improve this manuscript. 

Sincerely,

The autors

---

## [Decision Letter · Decision Letter 1]

16 Apr 2021

PONE-D-20-35553R1

Development of nano-emulsions based on Ayapana triplinervis essential oil for the control of Aedes aegypti larvae

PLOS ONE

Dear Dr. de Almeida,

Thank you for submitting your manuscript to PLOS ONE. After careful consideration, we feel that it has merit but does not fully meet PLOS ONE’s publication criteria as it currently stands. Therefore, we invite you to submit a revised version of the manuscript that addresses the points raised during the review process.

We look forward to receiving your revised manuscript.

Kind regards,

Ahmed Ibrahim Hasaballah

Academic Editor

PLOS ONE

Journal Requirements:

Reviewers' comments:

Reviewer's Responses to Questions

**Comments to the Author**

1. If the authors have adequately addressed your comments raised in a previous round of review and you feel that this manuscript is now acceptable for publication, you may indicate that here to bypass the “Comments to the Author” section, enter your conflict of interest statement in the “Confidential to Editor” section, and submit your "Accept" recommendation.

Reviewer #1: (No Response)

Reviewer #2: All comments have been addressed

2. Is the manuscript technically sound, and do the data support the conclusions?

Reviewer #1: Yes

Reviewer #2: Yes

3. Has the statistical analysis been performed appropriately and rigorously? 

Reviewer #1: Yes

Reviewer #2: Yes

4. Have the authors made all data underlying the findings in their manuscript fully available?

Reviewer #1: Yes

Reviewer #2: Yes

5. Is the manuscript presented in an intelligible fashion and written in standard English?

Reviewer #1: Yes

Reviewer #2: Yes

6. Review Comments to the Author

Reviewer #1: The authors may have not noted the concern by this reviewer with the mass spectra chromatograms that are included in the Supplementary Data as there was no author response to this point.

The concern noted in the review is here: "Notably, the Supplementary Data contains mass spectra chromatograms that appear to have been published elsewhere (e.g., by NIST and Adams (2017)). I doubt that PLOS permits the republishing of figures. I recommend the authors remove those figures and either replace them with mass spectra of standards that they generate for each chemical that was analyzed or state the diagnostic fragment ions of each chemical they analyze that match what is reported in the literature."

Please respond to the above comment and make any needed changes to the manuscript that are appropriate.

Authors did not include the scale bars in Fig. 4 that was recommended by the reviewer.

The other author responses to the reviewer comments are acceptable to this reviewer.

Reviewer #2: (No Response)

7. PLOS authors have the option to publish the peer review history of their article (what does this mean?). If published, this will include your full peer review and any attached files.

Reviewer #1: No

Reviewer #2: No

---

## [Author Response · Author response to Decision Letter 1]

17 Jun 2021

Ahmed Ibrahim Hasaballah

Academic Editor

PLOS ONE

We are writing to respond to reviewers for the requested adjustments to the manuscript entitled “PONE-D-20-35553R1 Development of nano-emulsions based on Ayapana triplinervis for the control of Aedes aegypti larvae”

JOURNAL REQUIREMENTS:

RESPONSE: The references were analyzed in detail according to the requirements of the scientific journal. The changes were highlighted in the manuscript, when necessary.

REVIEWER #1:

• The authors may have not noted the concern by this reviewer with the mass spectra chromatograms that are included in the Supplementary Data as there was no author response to this point.

The concern noted in the review is here: "Notably, the Supplementary Data contains mass spectra chromatograms that appear to have been published elsewhere (e.g., by NIST and Adams (2017)). I doubt that PLOS permits the republishing of figures. I recommend the authors remove those figures and either replace them with mass spectra of standards that they generate for each chemical that was analyzed or state the diagnostic fragment ions of each chemical they analyze that match what is reported in the literature." Please respond to the above comment and make any needed changes to the manuscript that are appropriate.

RESPONSE: We regret that this adjustment in the manuscript has gone unnoticed by our team. Receive our sincere apologies! We inform that the chromatograms were removed from the supplementary material, as instructed by the reviewer.

• Authors did not include the scale bars in Fig. 4 that was recommended by the reviewer.

RESPONSE: The scale bar was added to figure 4, according to the reviewer's guidance.

Additionally, we inform that the word nanoemulsion was standardized for nano-emulsion in the manuscript and in the supporting information.

Thank you for your time and consideration to improve this manuscript. 

Sincerely,

The autors

---

## [Editor Report · Decision Letter 2]

23 Jun 2021

Development of nano-emulsions based on Ayapana triplinervis essential oil for the control of Aedes aegypti larvae

PONE-D-20-35553R2

Dear Dr. de Almeida,

We’re pleased to inform you that your manuscript has been judged scientifically suitable for publication and will be formally accepted for publication once it meets all outstanding technical requirements.

Kind regards,

Ahmed Ibrahim Hasaballah

Academic Editor

PLOS ONE

---

## [Editor Report · Acceptance letter]

28 Jun 2021

PONE-D-20-35553R2 

Development of nano-emulsions based on *Ayapana triplinervis* essential oil for the control of *Aedes aegypti* larvae 

Dear Dr. de Almeida:

I'm pleased to inform you that your manuscript has been deemed suitable for publication in PLOS ONE. Congratulations! Your manuscript is now with our production department. 

Kind regards, 

on behalf of

Dr. Ahmed Ibrahim Hasaballah 

Academic Editor

PLOS ONE